# Prevalence, risk factors and predicted risk of cardiac events in chronic kidney disease of uncertain aetiology in Sri Lanka: A tubular interstitial nephropathy

Thilini W. Hettiarachchi[1]*, Buddhi N. T. W. Fernando[2], Thilini Sudeshika[3],
Zeid Badurdeen[1], Shuchi Anand[4], Ajith Kularatne[5], Sulochana Wijetunge[6], Hemalika T. K. Abeysundara[7], Nishantha Nanayakkara[8]

1 Centre for Education, Research and Training on Kidney Diseases (CERTKiD), Faculty of Medicine, University of Peradeniya, Galaha, Sri Lanka, 2 Department of Medical Laboratory Science, Faculty of Allied Health Sciences, University of Ruhuna, Matara, Sri Lanka, 3 Department of Pharmacy, Faculty of Allied Health Sciences, University of Peradeniya, Galaha, Sri Lanka, 4 Division of Nephrology, Stanford University School of Medicine, Stanford, California, United States of America, 5 Cardiology Unit, Teaching Hospital, Kandy, Sri Lanka, 6 Department of Pathology, Faculty of Medicine, University of Peradeniya, Galaha, Sri Lanka, 7 Department of Statistics and Computer Science, Faculty of Science, University of Peradeniya, Galaha, Sri Lanka, 8 Transplant and Dialysis Unit, Teaching Hospital, Kandy, Sri Lanka

* hatwasana@gmail.com

**Data Availability Statement:** All relevant data are within the paper.

## Abstract

Cardiovascular disease (CVD) is the leading cause of morbidity and mortality in patients with 'traditional' chronic kidney disease (CKD). However, chronic kidney disease of uncertain aetiology (CKDu), a tubular interstitial nephropathy is typically minimally proteinuric without high rates of associated hypertension or vascular disease and it is unknown if the rates of CVD are similar. This study aimed to identify the prevalence and the risk of CVD in patients with CKDu. This cross-sectional study included patients with confirmed CKDu who were attending two renal clinics in CKDu endemic-area. A detailed medical history, blood pressure, electrocardiogram (resting and six minutes vigorous walking), echocardiograms, appropriate laboratory parameters and medical record reviews were used to collect data at baseline. The WHO/Pan American Health Organization, cardiovascular risk calculator was employed to determine the future risk of CVD. The clinics had recorded 132 number of patients with CKDu, of these 119 consented to participation in the study. The mean age was 52 (± 9.5) years and mean eGFR was 51.1 (± 27.61); a majority (81.5% (n = 97)) were males. Thirty-four patients (28.6%) had evidence of ischaemic heart disease (IHD). Troponin-I (p = 0.02), Age >50 years (p = 0.01) and hyperuricemia (p = 0.01) were significantly associated with IHD in CKDu. Left ventricular hypertrophy was reported in 20.2% (n = 24). According to the risk calculator, 97% of the enrolled patients were at low risk (<10%) for experiencing a cardiovascular event within the next 10 years. Patients with CKDu have low prevalence and risk for CVD, implying that a majority are likely to survive to reach end-stage kidney disease. Our findings highlight the need for developing strategies to minimize the progression of CKDu to end-stage renal disease.

**Funding:** This study was funded by the National Research Council (Grant Number TO 14- 05), Sri Lanka The funders had no role in study design, data collection and analysis, decision to publish, or preparation of the manuscript.

**Competing interests:** The authors have declared that no competing interests exist.

## Introduction

Chronic kidney disease (CKD) is associated with an increased incidence of anaemia, bone mineral disease and cardiovascular disease (CVD) [1]. Moreover, CKD is regarded as an accelerator of CVD risk, and an independent risk factor for CVD events even in early stages of CKD [2, 3]. The spectrum of CVD in CKD includes ischaemic heart disease (IHD), left ventricular hypertrophy (LVH), vascular calcification, congestive heart failure, arrhythmias and peripheral vascular disease [4]. Multiple studies around the globe have reported a high prevalence of CVD among CKD patients. The incidence of CVD in CKD was 26.8%, 33.4%, 47.2%, and 39.1%, in CKD-ROUTE (Japan), CRIC (US), CRISIS (UK) and MERENA (Spain), respectively [5–8]. However, the prevalence of CVD in a recent Chinese cohort study (C-STRIDE) was low at 9.8% [9].

Even though patients at the early stages of CKD were absent of manifestations of vascular disease, they were associated with excess risk of subsequent coronary heart disease [10]. A large observational study of patients with early CKD followed for 5.5 years concluded that 24.9% of patients died before reaching dialysis and most of the deaths were attributed to cardiovascular events [11]. Thus, several guidelines recommend that the focus of patient care in the earlier stages of CKD should be directed at reducing cardiovascular mortality and morbidity [1].

CKD is strongly associated with increased cardiovascular morbidity and mortality independently from traditional cardiovascular risk factors (hypertension and diabetes). Novel risk factors of inflammation, bone and mineral disorders, hyperphosphatemia, hypercalcemia, secondary hyperparathyroidism, and oxidative stress, all of which are attributed to compromised renal function, are highly associated with elevated cardiovascular risk in patients with kidney disease [4].

Chronic kidney disease of uncertain aetiology (CKDu) is an endemic nephropathy, identified in distinct pockets in Sri Lanka, Nicaragua, El Salvador, Costa Rica, India and Egypt [12–14]. In Sri Lanka, CKDu was first recognized in late 1990 [13]. The male-dominant disease characteristically affects young or middle-aged farmers, who share underprivileged socio-economic backgrounds in rural dry zones of the country [15]. In practice, CKDu is commonly detected at community screening programs in already defined endemic zones [13]. Identified patients are further assessed at local renal clinics with routine renal profiles, urinalyses and ultrasound imaging of kidneys. Evolving criteria have been developed and implemented in the diagnosis of CKDu; based on growing evidence; fundamentally, it is a diagnosis by exclusion of known causes of CKD such as hypertension, diabetes vascular disease, glomerular nephritis, renal stone disease and other known renal diseases [16, 17]. Light microscopic findings in renal biopsies of CKDu are primarily tubular interstitial, and are characterized by interstitial fibrosis, tubular atrophy, glomerular sclerosis and periglomerular fibrosis [16]. Typically, immune-fluorescence is negative for IgG, IgM, and Complement-3 [15]. Rates of inadvertent exposure to agrochemicals [16, 18], poor nutritional status, smoking, betel chewing, recurrent dehydration and alcohol consumption [19] were shown to be higher and have been proposed as risk factors of in CKDu in affected populations by various groups. To the date, morbidity and detailed clinical characteristics have not properly articulated in this emerging disease of marginalized populations.

The risk of CVD may differ substantially in CKDu from other CKDs due to specific environmental, behavioural and disease characteristics. Understanding the importance, we designed this study to assess the burden of CVD, risk factors and the risk of developing CVD in CKDu, hypothesizing that the risk profile is different to CKD.

## Materials and methods

The cross-sectional descriptive study was conducted across two CKDu endemic regions of Sri Lanka, namely Wilgamuwa and Giradurukotte from July 2016 to end February 2017. Ethical clearance (Reference No: 2016/EC/29) was obtained from the Institutional Ethical Review Committee (IERC), Faculty of Medicine, University of Peradeniya, Sri Lanka. Patients followed up in the routine renal clinics at Girandurukotte and Wilgamuwa hospitals were informed regarding the research in their clinic visit. Thereafter, all patients with the diagnosis of confirmed CKDu [17] were invited to the study by medical officers of the clinics. Written informed consent was obtained from all recruited participants. The consent was documented in individual consent forms. The IERC approved the consent procedure.

### Patient questionnaire and medical record review

Investigators interviewed the participants and reviewed their medical records. The parameters ascertained were age, gender, IHD events, smoking status, blood pressure and body mass index (BMI), subsequent onset of diabetes and hypertension. Hypertension was defined as either systolic blood pressure >140 mmHg, or diastolic blood pressure >90 mmHg (at least three elevated readings, one week apart), or the current use of antihypertensive medications, or if self-reported. Diabetes was defined as HbA1c $\geq$ 6.5%, and/or initiation of insulin or oral hypoglycemic medications and/or if self-reported, after the diagnosis of CKDu.

### Laboratory parameters

Laboratory parameters were measured in a clinical laboratory at the regional tertiary care centre: Teaching Hospital, Kandy. Serum creatinine (Enzymatic method), total cholesterol by Cholesterol Oxidase (the single liquid method), calcium (Ca), phosphorous (P), uric acid were quantified by Indiko; Thermoscientific biochemical analyser, electrolytes (Ion-selective electrode method by Biolyte 3000), and haemoglobin (Hb) levels were measured using haematology analyser. Proteinuria was determined by sulfosalicylic acid test. Intact parathyroid hormone (iPTH), troponin I, high-sensitivity C-reactive protein (hsCRP) were measured using the chemiluminescence immunoassay analyser. Fibroblast growth factor 23 (FGF 23) and Cystatin C were determined using Luminex X- MAP technology detection of fluorescently labelled microsphere beads.

Clinical staging was determined by Kidney Disease Improving Global Outcomes (KDIGO) classification [20]. Estimated glomerular filtration rate (eGFR) was calculated using the Chronic Kidney Disease-Epidemiology Collaboration (CKD-EPI) equation [21]. Anaemia was defined as haemoglobin (Hb) concentration less than 13g/dL in males and less than 12g/dL in females [22] and hyperparathyroidism was defined as iPTH > 70pg/ml [23]. Hyperuricemia was defined as a serum uric acid level > 7.0 mg/dL in males and > 6.0 mg/dL in females [24].

### Cardiovascular disease evaluation

All patients underwent 12 lead electrocardiograms (ECG) at resting and after six minutes of vigorous walking. A cardiologist performed echocardiograms (2D Echo). Q waves or T waves abnormalities, bundle branch blocks, ST-segment changes on ECG and structural or functional cardiac abnormalities of 2D Echo were evaluated by an experienced cardiologist based on standard criteria (Table 1).

IHDs were classified under the categories of possible, probable or definite changes of ischaemia. Self-reported IHD (validated with discharge summary and hospital records), pathological Q waves or significant ST depressions or elevations either in resting or after exertion

**Table 1. Assessed abnormalities by ECG and echocardiography.**

| Diagnostic Tool | Abnormality | Definition |
|---|---|---|
| ECG | Pathological Q wave [25] | • Any Q-wave in leads V2–V3 ≥20 ms or QS complex in leads V2 and V3 |
| | | • Q-wave ≥30 ms and ≥0.1 mV deep or QS complex in leads I, II, aVL, aVF or V4–V6 in any 2 leads of a contiguous lead grouping (I, aVL, V6; V4–V6; II, III, and aVF) |
| | | • R-wave ≥40 ms in V1–V2 and R/S ≥1 with a concordant positive T-wave in the absence of a conduction defect |
| | ST depression [26] | • ST-segment depression of 1 mm or more 60–80 ms after the J point |
| | ST-elevation [27] | • Elevation of the ST segment at the J-point of above 0.2 mV in men 40 years of age or older, 0.25 mV or above in men below 40 years of age, and 0.15 mV or above in women in leads V2-V3 and/or 0.1 mV or above in all other leads |
| | T inversion [28] | • voltage of negative T-wave deep ≥ 5 mm in any of the leads |
| | Right bundle branch block (RBBB) | • Broad QRS > 120 ms |
| | | • RSR' pattern in V1-3 ('M-shaped' QRS complex) |
| | | • Wide, slurred S wave in the lateral leads (I, aVL, V5-6) |
| | Left bundle branch block (LBBB) | • QRS duration greater than 120 milliseconds |
| | | • Absence of Q wave in leads I, V5 and V6 |
| | | • Monomorphic R wave in I, V5 and V6 |
| | | • ST and T wave displacement opposite to the major |
| | | • deflection of the QRS complex |
| | Atrioventricular (AV) Block | • A PR interval consistently longer than 0.20 seconds |
| | Right axis deviation | • The QRS axis is shifted between 90 and 180 degrees |
| 2D Echocardiography | Regional wall motion abnormality (RWMA) | • Hypokinesis, dyskinesis, or akinesis of a segment when compared to the other contracting segments of the chamber. |
| | Left ventricular dialation [29] | • Left ventricular dialation >112% |
| | Left ventricular hypertrophy (LVH) | • Septal thickness of more than 1.5cm have been defined as LVH |

ECGs, RWMA in echocardiography were considered as definite IHD. Hospitalization for congestive heart failure, serious cardiac arrhythmia incidents (resuscitated cardiac arrest, ventricular fibrillation, sustained ventricular tachycardia, paroxysmal ventricular tachycardia, atrial fibrillation or flutter, severe bradycardia or heart block), LBBB, and left ventricular dilation were considered as probable IHD. Right axis deviation, atrial ventricular block, RBBB or T wave inversions in electrocardiography were classified as possible IHD.

### Definition of cardiovascular risk

There is no validated cardiovascular risk predictor for CKD/CKDu Sri Lanka. Pan American Health Organization/World Health Organization cardiovascular (PAHO/WHO) Risk Calculator was applied to calculate the cardiovascular risk in participants [30]. The rationale was the simplicity of the PAHO/WHO risk calculator and its use of readily available variables over other risk calculators (Framingham Risk Score, SCORE (Systematic Coronary Risk Evaluation) American College of Cardiology/American Heart Association Risk Model, 3rd Joint British Societies' risk score). The following parameters were included on the calculator gender, age, tobacco use (yes/no), systolic blood pressure, diabetes (yes/no), and total cholesterol level to predict the cardiovascular risk in the next 10 years. The risk stratification was set as follows; low risk: <10%, moderate: 10–19.9%, high: 20–29.9%, very high: 30–39.9%, and critical: ≥40% [30].

### Statistical analysis

Data were analysed via R statistical software. Baseline values were presented as mean ± standard deviation (SD) or medians and interquartile ranges for continuous variables,

and as numbers and percentages for categorical data. Baseline characteristics were compared between groups using the two-sample T-test, or two-sample proportion test, as appropriate. If the distribution of the continuous variable did not satisfy normal distribution, median values with interquartile ranges were reported and the Man-Whitney U test was used. A multivariable logistic regression model was employed to estimate the association between risk factors and IHD. All $p$-values were two-sided, and $p < 0.05$ was considered statistically significant.

## Results

Total CKD/CKDu patients registered at the two-referral centres (Wilgamuwa and Girandurukotte) was 2094. All participants with the diagnosis of confirmed CKDu (n = 132) were invited (Fig 1) and 119 with a mean duration of follow up of 6.2 ± 3.4 years were enrolled after written informed consent. The mean age of the participants was 52.0 (± 9.5) years; a majority (81.5% (n = 97) were men.

According to the CKD stage distribution 9.2% (11), 29.4% (35), 16.8% (20), 18.5% (22), 18.5% (22) and 7.6% (9) were at stage 1, 2, 3a, 3b, 4, and 5 respectively.

Majority of the participants (91.6%) were farmers with possible inadvertent, occupational exposure to agrochemicals. Smoking was not common (29.4% (n = 35)) among the participants. At the point of diagnosis, all the patients were non-hypertensive and reported with HbA1c ≤ 5.7% or fasting blood sugar < 6.9mmol/l, denoting that none of them were in prediabetes/diabetes conditions. Subsequent onset of hypertension and diabetes were observed in 25.2% (n = 30) and 17.7% (n = 21) respectively (Table 2). Hyperlipidaemia was observed in 22.7% (n = 27).

Proteinuria and age were significantly associated with lower eGFR <45 ml/min per 1.73 m². Even though the disease is not characterized by heavy proteinuria [31], the number of patients with proteinuria was significantly high in later stages (eGFR <45 ml/min per 1.73 m2) compared to the early stages.

Participants (25.2%, n = 30) were on antihypertensive medications (angiotensin-converting enzyme inhibitors, angiotensin receptor blockers, calcium blockers, alpha-blockers, beta-

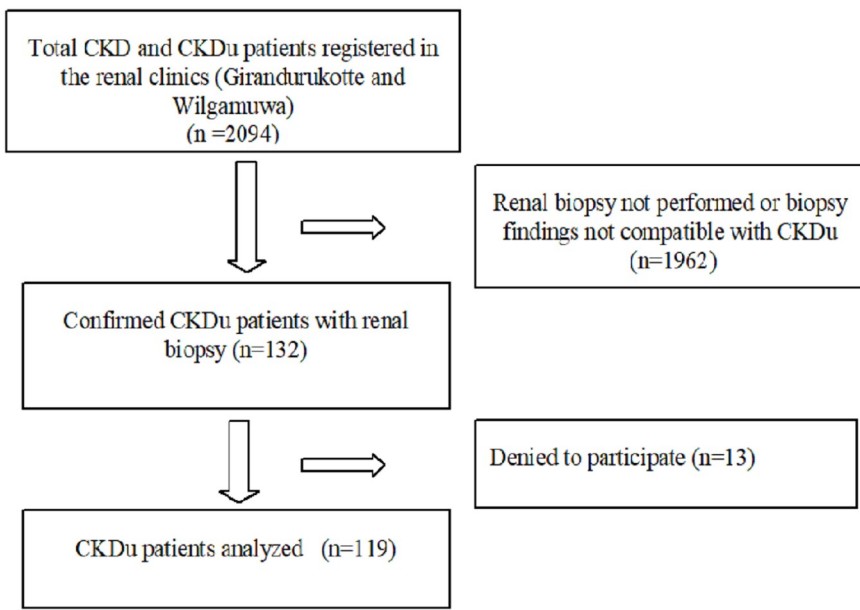

**Fig 1. Flow chart of the study enrolment.**

**Table 2. Baseline demographic characteristics and medical history of the study participants with eGFR and CKDu stages.**

| Demographic/clinical feature | Early Stage (stage 1- 3a), eGFR >45 ml/min per 1.73 m$^2$ | Late Stage (stage 3b- ESRD), eGFR <45 ml/min per 1.73 m$^2$ | p- Value |
|---|---|---|---|
| | N = 66 | N = 53 | |
| Mean Age (Mean ± SD) | 49.7 ± 9.19 | 55.0 ± 9.2 | 0.002* |
| Male (N, %) | 55 (83.3) | 42 (79.3) | 0.64 |
| Female (N, %) | 11 (16.7) | 11 (20.8) | 0.64 |
| BMI (kg/m2) (Mean ± SD) | 22.8 ± 3.1 | 22.1 ± 3.3 | 0.30 |
| Proteinuria (N, %) | 8 (12.1) | 23 (43.4) | < 0.001* |
| Subsequent onset of Hypertension (N, %) | 12 (18.2) | 18 (34) | 0.06 |
| Subsequent onset of Diabetes (N, %) | 10 (15.2) | 11 (20.8) | 0.47 |
| Hyperlipidaemia (N, %) | 17 (25.8) | 10 (18.9) | 0.39 |

eGFR -estimated Glomerular Filtration Rate, ESRD End-Stage Renal Disease, N-Number, BMI body mass index, Subsequent onset of hypertension and diabetes-hypertension and diabetes developed after the diagnosis of CKDu. Continuous variables are presented as mean ± SD, followed by two sample T-test, Categorical data are presented as numbers (n) of patients and percentages followed by two proportion test,

*Significance level p< 0.05.

blockers or diuretics) for a mean duration of 4.2 ± 2.6 years, 22.7% (n = 27) were on lipid-lowering therapy with statins for a mean period of 4.9 ± 3.1 years.

## Prevalence of cardiovascular diseases

**Ischaemic heart disease.** Thirty-nine coronary events were reported. Three patients were observed with two events in each (definite ischaemia and RBBB, possible ischaemia and right axis deviation, possible ischaemia and RBBB separately). One patient was observed with three events simultaneously (a history of myocardial infarction with possible ischaemia and RBBB in ECG. Only 34 (28.6%) patients had evidence of IHD, based on medical record review or measurements (Table 3).

From the patients with IHD, 32.4% (n = 11) was on antihypertensives and 23.5% (n = 8) was on statins. In the other group, who were without IHD, 22.4% (n = 19) was on antihypertensives and the percentage of patients on statins was 22.4% (n = 19) again. Notably, there was no significant difference between the proportion of patients on each treatment (antihypertensives (p = 0.35) or lipid-lowering agent; statin (p = 1.00)) in either group, with IHD or without IHD.

**Other CVDs in CKDu.** One patient had tachycardia (heart rate > 100 beats/min) and 31.1% (n = 37) had a heart rate less than 60 beats per minute (bpm) as per ECG evaluation. The mean heart rate was 67 (± 12.2) bpm. There was no incidence of atrial or ventricular tachycardia/fibrillation/flutter. Prolonged QRS complex (> 100 milliseconds) was observed in 29.4% (n = 35) of participants and it was positively correlated with IHD (p = 0.04). Mean corrected QT interval (QTc) for resting ECG was 414.5 (± 27.66)ms, and for the ECG after six minutes vigorous walking, it was 420.3 (± 34.29)ms. Six male patients (6.2%) and two (9.1%) female patients were observed with prolonged QTc interval (QTc interval > 450ms for males and QTc > 470 for females) in resting ECG.

Corresponding to the echocardiographic assessment, a majority (87.4%, n = 104) had left ventricular ejection fraction (LVEF) more than 60%. LVH was reported in 24 (20.2%) and only 5.88% (n = 7) had grade 1 diastolic dysfunction. There was no evidence of pulmonary hypertension or congenital abnormalities in 2D Echo. A normal pericardium was observed

**Table 3. IHD manifestations in CKDu Sri Lanka.**

| Classification of IHD | Specific cardiac abnormality (Total 119) | Number (%) |
|---|---|---|
| **Definite IHD** | Self-reported IHD; validated with discharge summary and hospital records | |
| | Myocardial infarction | 5 (4.2) |
| | History of acute coronary syndromes | 0 (0) |
| | History of coronary artery interventions | 0 (0) |
| | Pathological Q wave ST depression or elevation in ECG | 1 (0.8) |
| | T wave inversion with ST-segment depression | 13 (10.9) |
| | RWMA in Echocardiography | 0 (0) |
| | LBBB | 1 (0.8) |
| **Probable IHD** | Hospitalization for congestive heart failure | 0 (0) |
| | Serious cardiac arrhythmia incidents | 0 (0) |
| | Left ventricular dialation | 0 (0) |
| | Right axis deviation | 2 (1.7) |
| **Possible IHD** | RBBB | 15 (12.6) |
| | Atrioventricular (AV) Block | 2 (1.7) |

IHD- ischaemic heart disease, LBBB- Left bundle branch block, RBBB-Right bundle branch block, RWMA-Regional wall motion abnormalities.

No significant association was observed between the prevalence of IHD and CKDu stages.

in all the participants. Mitral, aortic or tricuspid regurgitation was not detected in echocardiograms.

X-ray images showed soft-tissue vascular calcification in six (5.04%) participants. Further, this was significantly associated (p = 0.003, α = 0.05) with lower eGFR.

## Cardiovascular risk prediction

**PAHO/WHO cardiovascular risk.** According to PAHO/WHO Cardiovascular Risk Calculator, one patient (0.8%) was in very high risk (30%-40%), two patients (1.7%) were in high risk (20%-30%) and again a single patient (0.8%) was at moderate risk (10%-20%) of developing cardiovascular events in the next ten years, while the majority (96.7%) were at low risk (<10%).

Figs 2 and 3 show the distribution of PAHO/WHO Cardiovascular Risk variables (total cholesterol, SBP age, diabetes and smoking) among different CKDu stages.

None of the patients had total cholesterol in cardiovascular risk level of greater than 240 mg/dl, and only eight (6.7%) were observed with SBP greater than the risk level of 140mmHg and those higher SBPs were from stage 5. Seventy patients (58.8%) were above the age of 50 years. (Fig 2).

The subsequent onset of diabetes was observed in 17.6% of CKDu patients with the incidence of 29.4% smoking. There was no significant difference in the proportions of diabetes and smoking across CKDu stages. Highest percentage (22.7%) of diabetes was observed in stage 3b. Stage 5 indicated the highest prevalence (44.4%) of smoking (Fig 3).

**Traditional and non-traditional risk factors.** Distribution of Cardiovascular risk factors and the use of medication among patients are shown in Table 4.

Hyperparathyroidism was observed in 27.7% (n = 33) of the study sample. A significantly higher proportion of patients with IHD group had hyperuricemia (p = 0.02) denoting that elevated serum uric acid level could be considered as a risk factor for IHD in CKDu. Even though the difference was not statistically significant, higher proportions of patients had nontraditional risk factors of CVD in the group of IHD.

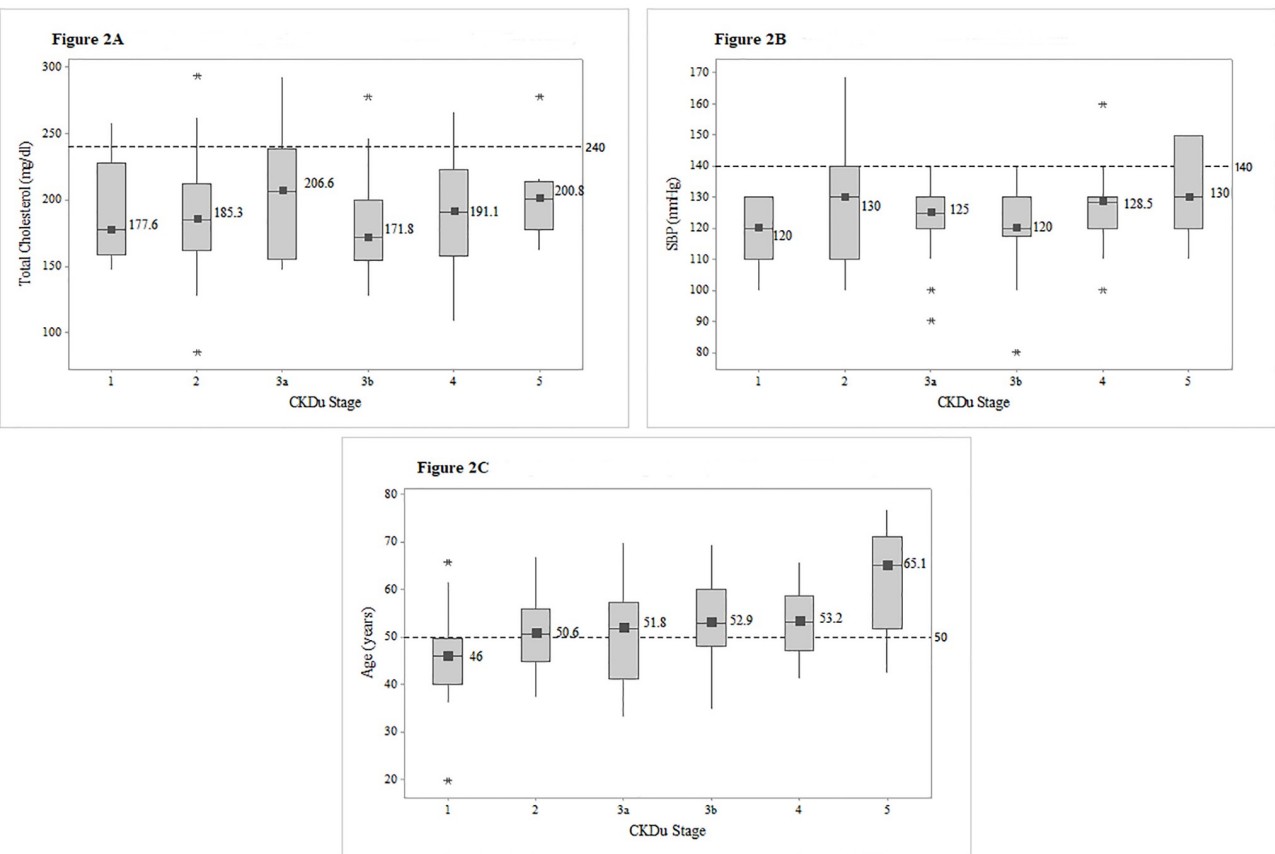

**Fig 2. Box plots of total cholesterol, SBP and age among different CKDu stages.** Distribution of total cholesterol (Fig 2A), systolic blood pressure (Fig 2B) and age (Fig 2C). among CKDu stages, with the median values for each stage. Reference lines were utilized to point out the proportion of patients exceeding the risk level of 240mg/dl in total cholesterol, 140mmHG in systolic blood pressure and 50 years in age. Abbreviations: SBP-Systolic blood pressure.

**Overall risk factors.** Troponin I level (p = 0.02), Age >50 years (p = 0.01) and hyperuricemia (p = 0.01) were significantly associated with the presence of IHD in multivariate analysis. Troponin I had a higher odd ratio of 7.16, indicating that every 1ng/ml increase in the troponin I level increases the risk of IHD by seven times.

Table 5 shows the results of the multiple logistic regression analysis for overall cardiovascular risk factors of IHD.

## Discussion

We report a lower prevalence of risk factors, evidence of CVD and low predicted risk of CVD in CKDu, a tubular interstitial disease. CKDu is a tubular interstitial nephropathy, which does not share common causes of CKD like diabetes, hypertension, glomerular nephritis [13, 32]. The disease is prevalent in remote dry zone of the country among socioeconomically disadvantaged populations. In general, CVD is considered as an important cause of morbidity and mortality in CKD. However, the prevalence of CVD risk factors and the impact of these risk factors on morbidity and mortality of CKDu have not yet been described.

Interestingly, there is no consensus among investigators on the criteria for the diagnosis of CVD in CKD. As a result, different criteria have been used to define CVD, making the comparison difficult. At least 35% of patients with CKD, had evidence of an ischaemic event

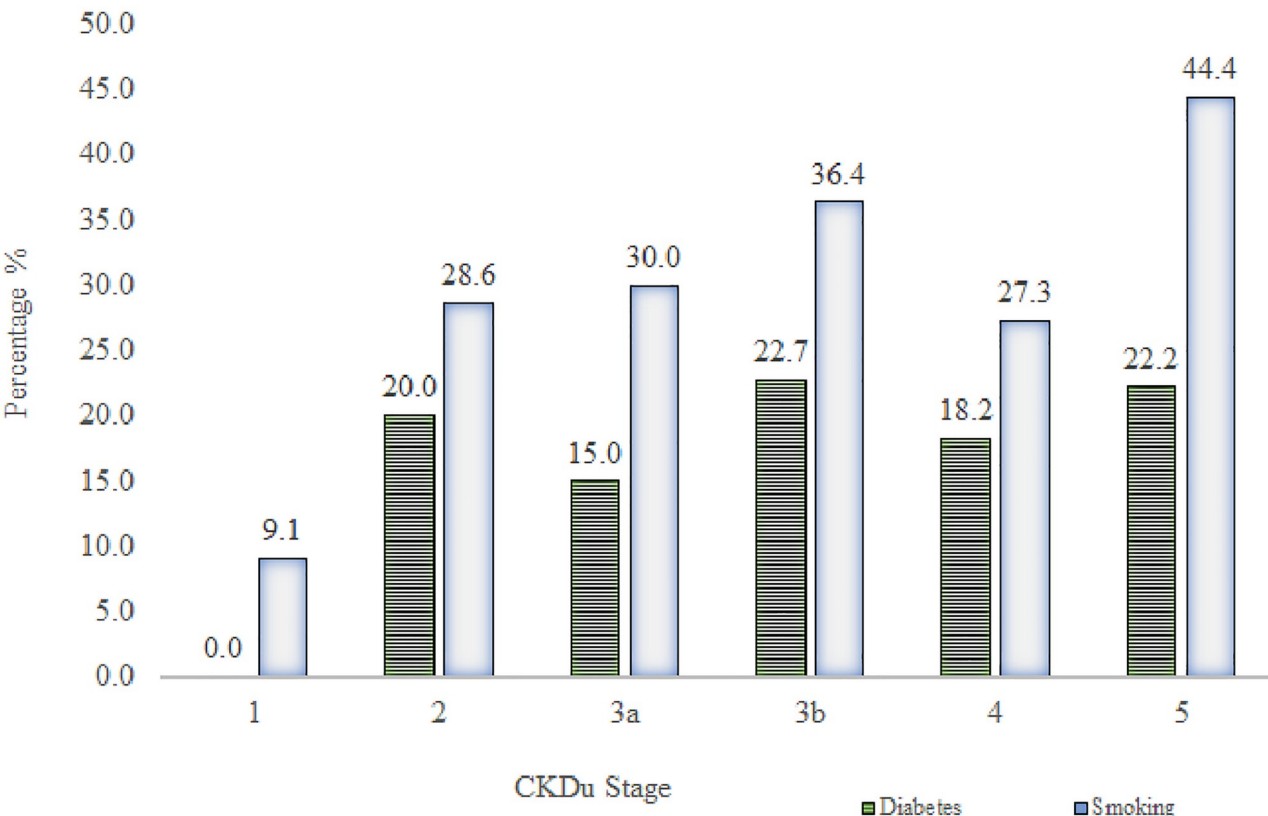

**Fig 3. Prevalence of diabetes and smoking in different CKDu cases.** Total bar height reflects the percentage of diabetes (HbA1c $\geq$ 6.5% or a history of diabetes, which was developed after the diagnosis of CKDu) and smoking in each stage of CKDu.

(myocardial infarction or angina) at the time of the presentation to a nephrologist [33]. In our group, the prevalence of ischaemic events was much lower, with 16% of definite 0.8% probable and 16% possible events. Furthermore, this finding is compatible with the characteristics of study population including physically active lifestyles, lower BMI, vegetable-based diet, lower prevalence of; diabetes, hypertension and proteinuria.

CKDu is a tubular interstitial disease among low socio-economic hardworking farmers in rural Sri Lanka [15]. IHD in CKD was multifactorial, and the incidence was decided by both general or traditional and disease-specific risk factors. In a Chinese cohort study, within the traditional risk factors, male gender, increasing age, smoking, established CVD, diabetes and increased total cholesterol was found to be associated with statistically significant increased risk of a cardiovascular event in CKD, while systolic and diastolic blood pressures were not associated with increased cardiovascular event risk [9]. In the instance of CKDu, though age >50 years was significantly associated with IHD, the other variables (male gender, smoking, total cholesterol) did not show any statistical significance. Similarly, there was no remarkable difference observed in IHD positive and negative group concerning BMI > 23kg/m$^2$ as seen in CKD [9, 34].

Apart from the above risk factors, in the current analysis of CKDu hyperuricemia and troponin-I showed a substantial risk for IHD. Hyperuricemia stimulates the renin-angiotensin system, and block the endothelial nitric oxide production, which contributes to renal vasoconstriction and hypertension. Ultimately, the hyperuricemia attributes to cardiovascular diseases [35, 36]. Moreover, hyperuricemia was independently associated with the development of IHD

**Table 4. Traditional and non-traditional cardiovascular risk factors in patients with and without IHD.**

| Variable | Patients with IHD (n = 34) | Patients without IHD (n = 85) | p-Value |
|---|---|---|---|
| Male | 31 (91.2) | 66 (77.6) | 0.12 |
| Female | 3 (8.8) | 19 (22.4) | 0.12 |
| **Traditional CVD Risk Factors** | | | |
| Age >50 years | 17 (50) | 50 (58.8) | 0.42 |
| BMI (kg/m2) > 23 | 15 (44.1) | 34 (40) | 0.69 |
| Smoking, current | 12 (35.3) | 23 (27.1) | 0.38 |
| HbA1c $\geq$ 5.7% | 16 (47.1) | 35 (41.2) | 0.68 |
| Mean total cholesterol (mg/dl) | 187.57 (± 38.8) | 179.15 (± 69.91) | 0.42 |
| Subsequent onset of Hypertension | 10 (29.4) | 20 (23.5) | 0.49 |
| Left ventricular hypertrophy | 8 (23.5) | 16 (18.8) | 0.62 |
| Vascular calcification | 2 (5.9) | 4 (4.7) | 1.00 |
| Proteinuria | 9 (26.5) | 22 (25.9) | 1.00 |
| **Non-traditional CVD risk factors** | | | |
| Hyperuricemia | 19 (55.9) | 28 (32.9) | 0.02* |
| Anaemia | 27 (79.4) | 59 (69.4) | 0.37 |
| eGFR (ml/min per 1.73 m$^2$) | 36 (40.25) | 53 (39) | 0.20 |
| Calcium (mg/dl) | 8.99 (± 0.5) | 9.15 (± 0.6) | 0.15 |
| Sodium (mmol/l) | 140 (7.15) | 142 (10.10) | 0.71 |
| Phosphate (mmol/l) | 1.015 (0.255) | 1.04 (0.25) | 0.92 |
| Potassium (mmol/l) | 4.598 (± 0.633) | 4.449 (±0.6082) | 0.25 |
| intact Parathyroid hormone (pg/ml) | 60.30 (34.21) | 54.85 (28.12) | 0.27 |
| high sensitivity C Reactive Protein (ng/ml) | 313 (1961) | 376 (1468) | 0.94 |
| Troponin I (ng/ml) | 0.015 (0.029) | 0.01 (0.023) | 1.00 |
| Serum bicarbonate (mmol/L) | 24.68 (±32) | 25.69 (±3.45) | 0.17 |
| Serum Cystatin C (ng/ml) | 2122 (1380) | 1865 (938) | 0.10 |
| Fibroblast growth factor (Pg/ml) | 23.68 (31.51) | 23.7 (22.3) | 0.66 |

Continuous variables are presented as mean ± SD, followed by two sample T-test and median (inter-quartile range) with the p-value from non-parametric Mann Whitney test. Categorical data are presented as numbers (n) of patients and percentages followed by two sample proportion test. IHD- Ischaemic heart Disease, BMI- body mass index, HbA1c- glycosylated haemoglobin type A1c, Subsequent onset of hypertension—hypertension developed after the diagnosis of CKDu, eGFR-estimated glomerular filtration rate,

*Significance level p< 0.05.

[37]. Prevalence of hyperuricemia was reported as high in CKDu in Central America [12, 38]. Similarly, in this study, we report a substantial-high percentage; around 40% with hyperuricemia in CKDu. Besides, according to our results, the presence of hyperuricemia was significantly associated with the prevalence of IHD (p = 0.02). Cardiac troponin-I is often elevated and predictive of mortality and cardiovascular events in CKD [39–41]. Correspondingly, we observed that elevated Troponin-I was positively correlated with IHD in CKDu.

Diabetes has been excluded in the diagnostic pipeline of CKDu. Nevertheless, 17.65%, (n = 21) had subsequent onset of diabetes or HbA1c $\geq$ 6.5% in the study, indicating that the two conditions can coexist commonly in Sri Lanka. In an epidemiological study conducted in an urban tertiary care centre of Sri Lanka found that the most common underlying cause of CKD was diabetes (n = 88, 44%) and again diabetes was the most common cause of CKD among patients from the western province (n = 74, 54%) [42]. However, previous studies have

**Table 5. Risk factors for the prevalence of IHD in CKDu.**

| Variable | Odds Ratio (95% CI) | *p*-Value |
|---|---|---|
| Male | 5.24 (0.68, 40.45) | 0.09 |
| **Traditional CVD Risk Factors** | | |
| Age >50 years | 0.17 (0.036, 0.77) | 0.01* |
| BMI (kg/m2) > 23 | 0.94 (0.28, 3.19) | 0.92 |
| Smoking, current | 1.65 (0.41, 6.70) | 0.48 |
| HbA1c ≥ 5.7% | 1.30 (0.36, 4.65) | 0.69 |
| Mean total cholesterol (mg/dl) | 1.005 (0.99, 1.02) | |
| Subsequent onset of Hypertension | 0.88 (0.16, 4.93) | 0.89 |
| Left ventricular hypertrophy | 1.71 (0.32, 9.31) | 0.54 |
| Vascular calcification | 4.42 (0.36, 53.78) | 0.25 |
| Proteinuria | 0.23 (0.042, 1.22) | 0.07 |
| **Non-traditional CVD Risk factors** | | |
| Hyperuricemia | 7.1 (1.38, 36.55) | 0.01* |
| Anaemia | 0.59 (0.12, 2.94) | 0.52 |
| eGFR (ml/min per 1.73 m$^2$) | 1.002 (0.97, 1.034) | 0.93 |
| Calcium (mg/dl) | 0.34 (0.096, 1.198) | 0.07 |
| Sodium (mmol/l) | 0.91 (0.80, 1.025) | 0.11 |
| Phosphate (mmol/l) | 5.02 (0.25, 99.91) | 0.27 |
| Potassium (mmol/l) | 2.14 (0.77, 5.98) | 0.13 |
| intact Parathyroid hormone (pg/ml) | 0.99 (0.98, 1.02) | 0.74 |
| high sensitivity C Reactive Protein (ng/ml) | 0.99 (0.99, 1.00) | 0.06 |
| Troponin I (ng/ml) | 7.16 (0.95, 54.13) | 0.02* |
| Serum bicarbonate (mmol/L) | 0.86 (0.68, 1.09) | 0.21 |

BMI- body mass index, HbA1c- glycosylated haemoglobin type A1c, Subsequent onset of hypertension—
hypertension developed after the diagnosis of CKDu, eGFR-estimated glomerular filtration rate,
*Significance level p< 0.05.

demonstrated that diabetes (2%, 9.6%) contributed to only a minority of CKD/CKDu in the North Central region of Sri Lanka [43, 44]. Renal impairment is commonly associated with glucose intolerance leading to Diabetes [45]. Notably, a previous study reported that HbA1C was significantly higher (p < 0.001) in CKD participants with IHD [9]. However, the current study did not show a similar association between IHD and HbA1c level. Incidence of smoking was not reported in the settings of CKD; Sri Lanka and in this study of Sri Lankan CKDu, it was 29.4%.

Proteinuria plays a significant role in the pathogenesis of IHD [46]. Nevertheless, in this study, the overall prevalence of proteinuria was as low as 26.05%. Hence, it was discovered that proteinuria was a late-stage finding in CKDu compared to early stages. Proteinuria has been associated with graded cardiovascular mortality, acting as risk multipliers across all levels of renal function in CKD [47, 48]. Contrasting to CKD, proteinuria does not seem to play a significant role in IHD pathogenesis in this specific disease of CKDu.

ECG is a widely available, time-tested simple investigation to identify cardiac diseases with significant precision. In this study, ECGs were used to identify rhythm abnormalities, ischaemic events and conduction defects. Kestenbaum *et al* reported that longer QRS intervals were independently associated with coronary artery disease in early CKD [49]. In the current analysis, prolonged QRS is positively associated with IHD, irrespective of the CKDu stage. Further,

in pre-dialysis settings, it has been described the QTc interval was significantly prolonged in CKD patients [49], but in our population similar prolongation was not observed.

LVH is the most common cardiac complication in CKD; a major risk factor for overall and cardiovascular mortality in patients with end-stage renal disease [50]. The incidence of LVH increased with the progressive decline in renal functions [51]. LVH in CKD is multifactorial and contributed by hypertension, anaemia and hyperparathyroidism [52]. However, limited data were available on the determinants of LVH in patients with CKDu. LVH in CKD is very common 70%- 80% [53]; nevertheless, the prevalence of LVH in the present study in CKDu is only 20.2% (24) of total participants. Lower prevalence of hypertension (25.2%), hyperparathyroidism (27.7%) in CKDu could be the reasons behind the lower prevalence of LVH.

Left ventricular (LV) diastolic dysfunction, frequently leading to congestive heart failure is common and almost universal in CKD [54]. In a study conducted in Tanzania, with 191 participants, recognized that the prevalence of LV diastolic dysfunction in CKD was 68.6% [55]. Nevertheless, only 5.88% of CKDu patients had LV diastolic dysfunction, and the prevalence was not different between stages. In the case of CKD, there was a stepwise reduction in LVEF with the progression of renal failure from stage 3 to stage 5 [56]. Contrary to that finding in CKD, there were no patients with lower LVEF less than 50% in the current study of CKDu.

The cardiovascular risk scores can estimate the probability of atherosclerotic cardiovascular events in patients with CKD regardless of renal function, albuminuria and previous cardiovascular events [57]. Herath *et al* in 2016 found that CVD risk assessment tools, both United Kingdom Prospective Diabetes Study risk engine and Framingham Risk Score have almost equal ability (former being marginally superior) in predicting underlying atherosclerotic vascular disease in patients with type 2 diabetes mellitus (T2DM) [58]. Nevertheless, in this study, we analysed a different population, who did not have a history of T2DM. Therefore, we used PAHO/WHO Risk Calculator, which is simple and easily implementable even in resource-poor settings with the readily available variables (gender, age, tobacco use, systolic blood pressure, diabetes and total cholesterol level) but with the limitation of not being validated for CKDu. The predicted cardiovascular risk score (3.4%) is remarkably low in CKDu, in comparison to higher cardiovascular risk (59%-75%) in CKD worldwide [9].

Proteinuria [59] and hypertension [60, 61] are considered as risk factors for CVD and mortality in patients with CKD. Hence, diabetes is associated with a higher risk for death [62]. Our population was observed with lower incidence of the above mortality risk factors of proteinuria (26.1%), subsequent hypertension (25.2%) and subsequent diabetes (17.6%). On top of that, these patients had a lower incidence of ischaemic heart disease. Conclusively, the results of the present study supported the lower prevalence of CVD events and lower predicted risk for CVD events in the population of patients with confirmed CKDu. It is implying that a majority are likely to survive to reach the end-stage kidney disease. Thus, this study suggests implementing strategies to minimize the progression is crucial in the management of CKDu.

The present study has several limitations. Since biopsy-proven/ confirmed CKDu cases were limited, we got relatively a small sample size of 119 from both the renal clinics. Hence, we may have missed a small group of participants with undiagnosed CKDu and for that reason; the results of CVD-related morbidity may not be all-inclusive. Hence, this is a cross-sectional study, which only utilized the data of a cross-section at a single time point. The other point is the utilised PAHO/WHO risk calculator was not validated to the CKDu Sri Lanka to predict the cardiovascular risk and which therefore may vary from the actual cardiovascular risk. Meanwhile, we agree a more appropriate validated risk prediction model is required for this particular group of patients for better results. The strength was the study population consisted exclusively of biopsy-proven confirmed CKDu patients in non-dialysis setting, which depict the authentic picture of the cardiovascular complications of CKDu in Sri Lanka.

## Acknowledgments

Authors are thankful to Dr Naradha Kodithuwakku at cardiology Unit, Teaching Hospital Kandy and Dr Lishantha Gunarathna and the staff of the Renal Centre at Girandurukotte, Sri Lanka.

## Author Contributions

**Conceptualization:** Thilini W. Hettiarachchi, Thilini Sudeshika, Zeid Badurdeen, Shuchi Anand, Ajith Kularatne, Sulochana Wijetunge, Hemalika T. K. Abeysundara, Nishantha Nanayakkara.

**Data curation:** Thilini W. Hettiarachchi, Buddhi N. T. W. Fernando, Thilini Sudeshika, Zeid Badurdeen, Shuchi Anand, Ajith Kularatne, Hemalika T. K. Abeysundara, Nishantha Nanayakkara.

**Formal analysis:** Thilini W. Hettiarachchi, Zeid Badurdeen, Shuchi Anand, Ajith Kularatne, Sulochana Wijetunge, Hemalika T. K. Abeysundara, Nishantha Nanayakkara.

**Investigation:** Thilini W. Hettiarachchi, Buddhi N. T. W. Fernando, Thilini Sudeshika, Zeid Badurdeen, Ajith Kularatne, Nishantha Nanayakkara.

**Methodology:** Thilini W. Hettiarachchi, Thilini Sudeshika, Zeid Badurdeen, Shuchi Anand, Ajith Kularatne, Sulochana Wijetunge, Nishantha Nanayakkara.

**Project administration:** Thilini W. Hettiarachchi, Buddhi N. T. W. Fernando, Thilini Sudeshika, Zeid Badurdeen, Ajith Kularatne, Sulochana Wijetunge, Nishantha Nanayakkara.

**Resources:** Thilini W. Hettiarachchi, Zeid Badurdeen, Shuchi Anand, Nishantha Nanayakkara.

**Software:** Thilini W. Hettiarachchi, Hemalika T. K. Abeysundara.

**Supervision:** Thilini W. Hettiarachchi, Thilini Sudeshika, Zeid Badurdeen, Shuchi Anand, Ajith Kularatne, Sulochana Wijetunge, Hemalika T. K. Abeysundara, Nishantha Nanayakkara.

**Validation:** Thilini W. Hettiarachchi, Zeid Badurdeen, Shuchi Anand, Ajith Kularatne, Sulochana Wijetunge, Hemalika T. K. Abeysundara, Nishantha Nanayakkara.

**Visualization:** Thilini W. Hettiarachchi, Shuchi Anand, Ajith Kularatne.

**Writing – original draft:** Thilini W. Hettiarachchi, Shuchi Anand, Nishantha Nanayakkara.

**Writing – review & editing:** Thilini W. Hettiarachchi, Buddhi N. T. W. Fernando, Thilini Sudeshika, Zeid Badurdeen, Shuchi Anand, Ajith Kularatne, Sulochana Wijetunge, Hemalika T. K. Abeysundara, Nishantha Nanayakkara.

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
