## [Decision Letter · Decision Letter 0]

28 Sep 2020

PONE-D-20-19631

Cardiovascular morbidity in patients with Chronic Kidney Disease of uncertain aetiology in Sri Lanka: a tubular interstitial nephropathy

PLOS ONE

Dear Dr. Hettiarachchi,

Thank you for submitting your manuscript to PLOS ONE. After careful consideration, we feel that it has merit but does not fully meet PLOS ONE’s publication criteria as it currently stands. Therefore, we invite you to submit a revised version of the manuscript that addresses the points raised during the review process.

The reviewers have raised important issues which need to be addressed to strengthen the paper. 

We look forward to receiving your revised manuscript.

Kind regards,

Rohina Joshi

Academic Editor

PLOS ONE

Journal Requirements:

a) Did participants provide their written or verbal informed consent to participate in this study?

Reviewers' comments:

Reviewer's Responses to Questions

**Comments to the Author**

1. Is the manuscript technically sound, and do the data support the conclusions?

Reviewer #1: Partly

Reviewer #2: Yes

2. Has the statistical analysis been performed appropriately and rigorously? 

Reviewer #1: Yes

Reviewer #2: Yes

3. Have the authors made all data underlying the findings in their manuscript fully available?

Reviewer #1: Yes

Reviewer #2: No

4. Is the manuscript presented in an intelligible fashion and written in standard English?

Reviewer #1: Yes

Reviewer #2: Yes

5. Review Comments to the Author

Reviewer #1: The study addresses an important yet unaddressed topic - Cardiovascular risk in CKDu.

The authors outline the study aims as identifying the prevalence and the risk of CVD in patients with CKDu, however the methods described in the study and sampling approaches would need to address the following

Mean duration of hospital based follow up of patients with CKDu at the study sites as this are likely to have an impact on the findings on the cardiovascular risk profile.

Sample size calculations ( taking into consideration the prevalence of CKDu in the endemic regions of Wilgamuwa and Giradurukotte).

Any attempts to document history of first degree relatives with cardiovascular events or inherited disorders of lipid metabolism ( Familial Hypercholesterolemia).

Cardiovascular risk calculation - Prior studies from Srilanka have indicated that Framingham Risk / UKPS risk engines are better for CVD risk screening for detecting subclinical Atherosclerosis as compared to the WHO Risk tools (Journal of Clinical and Diagnostic Research: JCDR. 2016 Jul;10(7):OC09.)

More details on the following would be useful

Line 164: All 132 CKDu confirmed subjects - Where these newly confirmed CKDu during the study period at these two referral centres?. It would also be useful to understand the proportion of CKD patient pool from which these were confirmed and if there were any similarities in the baseline risk profiles as the manuscript restricts the analysis to CKDu with or without IHD however draws inferences comparing with CKD in general.

Line 193 Table 2 : Self reported IHD, Myocardial Infarction in 5 participants - was this validated with hospital records, discharge summary or EKG/ ECHO findings during the evaluation.

Line 336: Hyperuricemia showed a substantial risk for IHD. There have been reports on the potential association of Hyperuricemia as a risk for CKDu in the Mesoamerican Nephropathy ( Line 341) suggesting that repeated episodes of AKI might be resulting in CKDu. Is the study adequately powered to draw the inference the significant association?

Line 348 17.65%,(n=21) of later onset of diabetes or HbA1c ≥ 6.5% was identified, would be important to establish what proportion of the subjects had HbA1C values qualifying for Prediabetes at the time of CKDu diagnosis?

Line 353 The study did not show a similar association between IHD and HbA1c level,

Neither does the study show any association with respect to an increased BMI between those with IHD in the CKDu and those without IHD ( Table 2 , BMI > 23 p =0.686) , it might be important highlight if there are any such differences in among the CKD populations with or without IHD in the Sri Lankan context.

It would be important to also know what proportion of the subjects were on ACE inhibitors, angiotensin receptor blockers, ß blockers, ASA and Statins and duration of being on these as the cardiovascular risk assessments are likely to be impacted by these interventions.

Reviewer #2: PLOS ONE

Cardiovascular morbidity in patients with Chronic Kidney Disease of uncertain aetiology in Sri Lanka: a tubular interstitial nephropathy

This is an interesting piece of work and relevant for clinicians globally caring for patients with CKDu. The question of whether the risk of cardiovascular disease/morbidity in patients with CKDu is the same as those with CKD is important for patients, clinicians and health care providers.

The authors have conducted a study with reasonable scientific rigour, but the writing needs some more work to make the results easy to understand.

The discussion needs to be reframed to focus on the main finding that the rates of cv events are lower and patients are likely to survive to needing treatment for eskd and delaying the onset of eskd is crucial to their management

Please see specific comments below.

Abstract

Line 30 – I suggest re-write as below

Cardiovascular disease (CVD) is the leading cause of morbidity and mortality in patients with ‘traditional’ chronic kidney disease (CKD), however, chronic kidney disease of uncertain aetiology (CKDu), a tubular interstitial nephropathy is typically minimally proteinuric without high rates of associated hypertension or vascular disease and it is unknown if the rates of cardiovascular disease are similar.

Line 33 - This study aimed to identify the prevalence and the risk of CVD in patients with CKDu. Delete the rest of the sentence.

Line 36 remove the word ‘to.’ Instead of the work current use baseline

Perhaps rewrite the sentence starting with in order to –

A detailed medical history, blood pressure, electrocardiogram (resting and six minutes vigorous walking) and echocardiograms, appropriate laboratory parameters and a medical record review was used to collect data at baseline. The WHO/Pan American Health Organization, cardiovascular risk calculator was employed to determine the future risk of CVD.

Line 41 – rewrite to –

The clinics had recorded xx number of patients with CKDu, of these 119 consented to participation in the study.

Line 44 – add years

Line 49 – add planning for eskd services or something along those lines.

Introduction

Line 55 – Perhaps say tat all stages of CKD are associated with increased cvd risk

Line 57 - ?delete the sentence starting with structural and ….

Line 63/Paragraph 2 – should discuss the international findings a little more so its more relevant to a global audience such as PLOS one.

Line 84 – the sentence starting – in order to …. Is methods – please deelte

Line 90 -across rather than in

How were patietns invited? Need to say somewhere that they were consented and by who.

Line 102 – what do you mean by later onset of diabetes? Subsequent perhaps?

Line 122 – reorder the sentence starting with according to kdigo,

Could just say that the KDIGO definitions were use for stage of ckd and anemia. And reference to a table with all the definitions in the table. Will be easier to read

Line 129 – who assessed the ECGs and was this standardised?

Who performed the echo’s and was any sort of standard criteria used to record abnormalities?

Results

Needs a study flow diagram that shows how many people were invited, how many consented.

Line 172 can be removed and table 1 in brackets can be added at the end of the first sentence. Was age normally distributed? If not, best to report medians.

Line 181 is a key finding and should be given more prominence. It suggests that proteinuria is a negative marker even in a disease not characterised by proteinuria.

Line 185 – please re-write, I found it very difficult to understand. I also didn’t understand if the 1 patient (line 187) was from the 119 or out of the 39.

Table 2 – need a total at the top – was this out of 119 patients or 39 with abnormalities

Line 201 – might be easier to say no incidence of …..

Regarding the figures – it might be helpful for the readers to have some context in terms of the rates of smoking and diabetes in Sri Lanka or among ‘traditional’ CKD patients.

It might be easier to describe this as a group of rhythm abnormalities – easier to read and understand

Line 242 – important point – move up;

Table 3 Best to avoid abbreviations as much as possible such as LVH, K, P, Na, Ca HCO3 – use the words instead

Line 272 – instead of overall risk factor better to call it risk calculator or something like that

Start the paragraph with line 281 rather than the tables.

My earlier comments about abbreviations apply to all the tables.

The discussion needs to re-written so it is focussed on the main findings of the study. Every finding of the study does not need to be discussed. The key finding is that patients with CKDu have lower risk fo CVD. There is no mention of the outcome of mortality anywhere and that would be useful to discuss and report on in this group. The first 3 paragraphs especially should be re-written.

Line 325 – is not was

Line 334 – How common is BMI>24 in SriLanka?

Line 356 proteinuria is an important finding and should be given prominence

The limitations and strengths should be addressed openly.

Line 365 - please rewrite. I did not understand

6. PLOS authors have the option to publish the peer review history of their article (what does this mean?). If published, this will include your full peer review and any attached files.

Reviewer #1: No

Reviewer #2: **Yes: **Sradha S Kotwal

---

## [Author Response · Author response to Decision Letter 0]

10 Nov 2020

Journal Requirements:

 We appreciate this feedback from the Editors.

Comment 1: Please ensure that your manuscript meets PLOS ONE's style requirements, including those for file naming. The PLOS ONE style templates can be found at

Response: 

Complied with the given formats.

Comment 2: Please amend your current ethics statement to address the following concerns: 

a) Did participants provide their written or verbal informed consent to participate in this study? 

Response: 

Please find the amended ethics statement (lines 106-108).

a). Yes, they have provided written informed consent b). i) No, the consent was not verbal, it is written. ii) Consent was documented in individual consent forms. iii). Yes, ethics committee approved this consent procedure. 

 Review Comments to the Author

Reviewer #1: The study addresses an important yet unaddressed topic - Cardiovascular risk in CKDu. The authors outline the study aims as identifying the prevalence and the risk of CVD in patients with CKDu, However, the methods described in the study and sampling approaches would need to address the following.

We were highly impressed by the statement on understanding the importance of this sudsy in an area with limited knowledge.

Comment 1: Mean duration of hospital-based follow up of patients with CKDu at the study sites as this are likely to have an impact on the findings on the cardiovascular risk profile.

Response: As reviewer correctly identified these patients were regularly followed up at two renal clinics in the endemic area of CKDu, Sri Lanka. Mean duration of the hospital-based follow up was 6.2 ± 3.4 years (mentioned in the line 182).

Comment 2: Sample size calculations (taking into consideration the prevalence of CKDu in the endemic regions of Wilgamuwa and Giradurukotte).

Response: We have invited all the followed-up patients (n=132) with confirmed CKDu (renal biopsy findings compatible) at renal clinic Wilgamuwa and Girandurukotte during the period from July 2016 to end February 2017. Thirteen patients denied the participation. All remaining participants were included as in total purposive sampling. Since all the available patient count is less as 119, no special randomization technique or sample size calculation was used.

Comment 3: Any attempts to document history of first-degree relatives with cardiovascular events or inherited disorders of lipid metabolism (Familial Hypercholesterolemia).

Response: We agree with the reviewer importance of familial risk factors of CVD, but unfortunately, we have not documented the history of first-degree relatives with cardiovascular events or dyslipidaemias

Comment 4: Cardiovascular risk calculation - Prior studies from Sri Lanka have indicated that Framingham Risk / UKPS risk engines are better for CVD risk screening for detecting subclinical Atherosclerosis as compared to the WHO Risk tools (Journal of Clinical and Diagnostic Research: JCDR. 2016 Jul;10(7): OC09.)

Response: We also had similar concerns in selecting appropriate prediction model. Herath et al in 2016 found that CVD risk assessment tools, both United Kingdom Prospective Diabetes Study risk engine and Framingham Risk Score have almost equal ability (former being marginally superior) in predicting underlying atherosclerotic vascular disease in patients with type 2 diabetes mellitus (T2DM). Nevertheless, in this study, we analysed a different population of patients with CKDu, who did not have a history of type 2 diabetes mellitus. Hence, there is no specific tool for cardiovascular risk prediction in CKDu Sri Lanka we used PAHO/WHO (Pan American Health Organization/World Health Organization cardiovascular) Risk Calculator in predicting cardiovascular risk in this specific population, which is more simpler and easily implementable even in resource-poor settings. Meanwhile, we agree a correct validated risk prediction model is required for each specific group of patients for better results.

(Herath H. M., Weerarathna T. P., Dulanjalee R. B., Jayawardana M. R., Edirisingha U. P., Rathnayake M. Association of Risk Estimates of Three Different Cardiovascular Risk Assessment Tools with Carotid Intima Media Thickness in Patients with Type 2 Diabetes. J. Clin. Diag. Res. 2016;10(7) OC09-12)

More details on the following would be useful

Comment 5: Line 164: All 132 CKDu confirmed subjects - Where these newly confirmed CKDu during the study period at these two referral centres?. It would also be useful to understand the proportion of CKD patient pool from which these were confirmed and if there were any similarities in the baseline risk profiles as the manuscript restricts the analysis to CKDu with or without IHD however draws inferences comparing with CKD in general.

Response: Total CKD/CKDu patients registered at the two-referral centres (Wilgamuwa and Girandurukotte) was 2094. Only 132 patients were observed with confirmed CKDu; which was determined through a kidney biopsy. (Clarified in lines 180-181 and fig 1). Histopathological features consistent with CKDu on biopsy are the features of (a) Glomerular sclerosis and glomerular collapse, (b) interstitial fibrosis, (c) tubular fibrosis, (d) tubular atrophy, (e) interstitial infiltration, (f) presence of casts, and (h) arteriolar hyalinosis (Selvarajah et al., 2016), preferably the demonstration of the absence of immune deposits. Yet, there may be other patients with CKDu; who had denied undergoing biopsy procedures. Therefore, it was not practical to demarcate CKD and confirmed CKDu without kidney biopsy. In that case, we only considered patients with confirmed CKDu, to obtain a better clinical picture of the cardiovascular morbidity in patients with CKDu. Since there is no literature on CKDu related cardiovascular morbidity, we drew inferences comparing with CKD in general. It is useful to compare the baselines of different groups within CKD, but unfortunately, it was not studied and a limitation to generalize our findings.

(Selvarajah M, Weeratunga P, Sivayoganthan S, Rathnatunga N, Rajapakse S. Clinicopathological correlates of chronic kidney disease of unknown etiology in Sri Lanka. Indian J Nephrol. 2016;26(5): 357–363. pmid:27795631)

Comment 6: Line 193 Table 2: Self-reported IHD, Myocardial Infarction in 5 participants - was this validated with hospital records, discharge summary or EKG/ ECHO findings during the evaluation.

Response: Yes, all the events of myocardial infarction was validated with discharge summary and hospital records. (Mentioned in the top row in Table 3)

Comment 7; Line 336: Hyperuricemia showed a substantial risk for IHD. There have been reports on the potential association of Hyperuricemia as a risk for CKDu in the Mesoamerican Nephropathy( Line 341) suggesting that repeated episodes of AKI might be resulting in CKDu. Is the study adequately powered to draw the inference the significant association?

Response: We also think hyperuricaemia is an important molecule in the pathogenesis and other manifestations of CKDu. We have stated that “Prevalence of hyperuricemia was reported as high in CKDu in Central American studies of Mesoamerican Nephropathy. Similarly in this study, we report a substantial-high percentage; around 40% with hyperuricemia. In addition to that, the presentation of hyperuricemia in this study of Sri Lankan CKDu has a significant association with IHD”. The study is not adequately powered to comment on AKI events. Revised the lines 356-359

Comment 8: Line 348 17.65%,(n=21) of later onset of diabetes or HbA1c ≥ 6.5% was identified, would be important to establish what proportion of the subjects had HbA1C values qualifying for Prediabetes at the time of CKDu diagnosis?

Response: At the time of diagnosis, all the patients were reported as not having Diabetes Mellitus either by HbA1c < 6.4% or fasting blood sugar less than 6.9mmol/l, denoting that none of the patients was Diabetic at the time of diagnosis as per MOH diagnostic criteria(explained in lines 192-193, 363). Even though it is very useful to know, the number who were in the prediabetes range at the time of diagnosis was difficult to extract from available records.

Comment 9: Line 353, The study did not show a similar association between IHD and HbA1c level, neither does the study show any association with respect to an increased BMI between those with IHD in the CKDu and those without IHD ( Table 2, BMI > 23 p =0.686). It might be important to highlight if there are any such differences in among the CKD populations with or without IHD in the Sri Lankan context.

Response: In the current study of CKDu we did not observe any significant association between IHD and HbA1c level and again IHD and BMI (kg/m2) > 23. We admit that it is much informative if we compare IHD and HbA1c and IHD and BMI in both CKD and CKDu in Sri Lanka. However, there are no reported literature on HbA1c levels or BMI among the CKD populations with or without IHD in the Sri Lankan context. Therefore, such comparisons were impossible in the current study.

Comment 10: It would be important to also know what proportion of the subjects were on ACE inhibitors, angiotensin receptor blockers, ß blockers, ASA and Statins and duration of being on these as the cardiovascular risk assessments are likely to be impacted by these interventions.

Response: We have analysed the medications and duration of being on those medications (Explained in the lines 211-214). Of note, there was no significant impact from the medications in the current study (described in lines 230-234). 

“Participants (25.2%, n=30) were on antihypertensive medications (angiotensin-converting enzyme inhibitors, angiotensin receptor blockers, calcium blockers, alpha-blockers, beta-blockers or diuretics) for a mean duration of 4.2 ± 2.6 years, 22.7% (n=27) were on lipid-lowering therapy with statins for a mean period of 4.9 ± 3.1 years.

From the patients with IHD, 32.4% (n=11) was on antihypertensives and 23.5% (n=8) was on statins. In the other group, who were without IHD, 22.4% (n=19) on antihypertensives and 22.4% (n=19) on statins. Notably, there was no significant difference between the proportion of patients on each treatment (antihypertensives (p= 0.350) or lipid-lowering agent; statin (p =1.000)) in either group, with IHD or without IHD.”

Reviewer #2: PLOS ONE

This is an interesting piece of work and relevant for clinicians globally caring for patients with CKDu. The question of whether the risk of cardiovascular disease/morbidity in patients with CKDu is the same as those with CKD is important for patients, clinicians and health care providers.

The authors have conducted a study with reasonable scientific rigour, but the writing needs some more work to make the results easy to understand.

The discussion needs to be reframed to focus on the main finding that the rates of cv events are lower and patients are likely to survive to needing treatment for eskd and delaying the onset of eskd is crucial to their management

We highly acknowledge the comments from the reviewer #2. We have restructured the discussion by focusing on the main findings that rates of CV events are lower and concluding that patients are likely to survive to ESKD and delaying the onset of ESKD is crucial. 

Please see specific comments below.

Comment 1: Abstract: Line 30 – I suggest re-write as below; Cardiovascular disease (CVD) is the leading cause of morbidity and mortality in patients with ‘traditional’ chronic kidney disease (CKD. However, chronic kidney disease of uncertain aetiology (CKDu), a tubular interstitial nephropathy is typically minimally proteinuric without high rates of associated hypertension or vascular disease and it is unknown if the rates of cardiovascular disease are similar.

Response: Admitted and corrected in the lines 29-33

Comment 2: Line 33 - This study aimed to identify the prevalence and the risk of CVD in patients with CKDu. Delete the rest of the sentence.

Response: Admitted and corrected in the lines 33-34

Comment 3: Line 36 remove the word ‘to.’ Instead of the work current use baseline

Perhaps rewrite the sentence starting with in order to –

A detailed medical history, blood pressure, electrocardiogram (resting and six minutes vigorous walking) and echocardiograms, appropriate laboratory parameters and a medical record review was used to collect data at baseline. The WHO/Pan American Health Organization, cardiovascular risk calculator was employed to determine the future risk of CVD.

Response: Admitted and corrected in the lines 36-40

Comment 4: Line 41 – rewrite to –

The clinics had recorded xx number of patients with CKDu, of these 119 consented to participation in the study.

Response: Admitted and corrected in the lines 41-42

Comment 5: Line 44 – add years

Response: Done, in the line 44

Comment 6: Line 49 – add planning for eskd services or something along those lines.

Response: Corrected as “Our findings highlight the need for developing strategies to minimize the progression of CKDu to end-stage renal disease” (lines 50-51)

Introduction

Comment 7; line 55 – Perhaps say that all stages of CKD are associated with increased cvd risk

Response: We wanted to elaborate that, even early stages of CKD are associated with increased CVD risk. So we updated the sentence in the lines 54-55.

Comment 8; Line 57 - ?delete the sentence starting with structural and ….

Response: Admitted and deleted

Comment 9: Line 63/Paragraph 2 – should discuss the international findings a little more so its’ more relevant to a global audience such as PLOS one.

Response: Admitted with thanks. We have discussed the international findings in the lines 58-76

Comment 10; Line 84 – the sentence starting – in order to …. Is methods – please delete

Response: Admitted and deleted 

Materials and methods

Comment 11; Line 90 -across rather than in

Response: Corrected ;line 102

Comment 12: How were patients invited? Need to say somewhere that they were consented and by who.

Response: Patients were recruited on written informed consent by trained researchers (mentioned in the lines 105-107)

Comment 13: Line 102 – what do you mean by later onset of diabetes? Subsequent perhaps?

Response: Yes, Changed the word to subsequent onset of diabetes

Comment 14: Line 122 – reorder the sentence starting with according to kdigo,

Could just say that the KDIGO definitions were used for stage of ckd and anemia. And reference to a table with all the definitions in the table. Will be easier to read

Response: Changed the sentence and updated the overall paragraph with definitions for anaemia, hyperparathyroidism and hyperuricemia (in the lines 128-133)

Comment 15: Line 129 – who assessed the ECGs and was this standardised?

Response: a cardiologist assessed ECG, based on standard criteria for ECG abnormalities (explained in the lines 135-139 and Table 1). (If the table 1 is over informative, we would like to convert that to a paragraph)

Comment 16: Who performed the echo’s and was any sort of standard criteria used to record abnormalities?

Response: An experienced cardiologist performed Echocardiograms and standard criteria was used to record the abnormalities (explained in the lines 135-139 and Table 1). (If the table 1 is over informative, we would like to convert that to a paragraph)

Results

Comment 17: Needs a study flow diagram that shows how many people were invited, how many consented.

Response: The total CKD/CKDu patients registered at both clinics was 2094. Among them, all confirmed CKDu subjects (132) were invited and 119 were consented to participate in the study. (Flow chart inserted as Fig 1, described in the lines 180-182). 

Comment 18: Line 172 can be removed and table 1 in brackets can be added at the end of the first sentence. Was age normally distributed? If not, best to report medians.

Response: The line was removed and Table 2 (previous table 1) was cited within brackets in the text (line 195). Age was normally distributed. Therefore, we used mean and standard deviation.

Comment 19: Line 181 is a key finding and should be given more prominence. It suggests that proteinuria is a negative marker even in a disease not characterised by proteinuria.

Response: Even the disease is not characterized by heavy proteinuria; number of patients with proteinuria was significantly high in later stages compared to the early stages. (Described in the lines 207-209)

Comment 20: Line 185 – please re-write, I found it very difficult to understand. I also didn’t understand if the 1 patient (line 187) was from the 119 or out of the 39.

Response: We have rewritten the paragraph (lines 220-223)

Comment 21: Table 2 – need a total at the top – was this out of 119 patients or 39 with abnormalities

Response: Added the total number on top of the table, it is from overall 119 

Comment 22: Line 201 – might be easier to say no incidence of …..

Response: Corrected in the line 238

Comment 23: Regarding the figures – it might be helpful for the readers to have some context in terms of the rates of smoking and diabetes in Sri Lanka or among ‘traditional’ CKD patients.

Response: An epidemiology study carried-out in urban tertiary care centre of Sri Lanka found that the most common underlying causes of CKD were diabetes (88, 44%) and again diabetes was the most common cause of CKD among patients from the western province (74, 54%) (Wijewickrama et al., 2011). However, previous studies have demonstrated that diabetes (2%, 9.6%) contributed to only a minority of CKD/CKDu in the North Central region of Sri Lanka [Athuraliya et al., 2009 and Katulanda et al., 2011]. In the current study of the two CKDu endemic areas of the North Central region observed 17.6% of CKDu patients with the subsequent onset of diabetes. Incidence of smoking was not reported in the settings of CKD, Sri Lanka and the current study of Sri Lankan CKDu, it is 29.4% (Explained in the lines 364-369 and 372-374)

(Wijewickrama ES, Weerasinghe D, Sumathipala PS, Horadagoda C, Lanarolle RD, Sheriff RMH. Epidemiologyof Chronic Kidney Disease in a Sri Lankan Population:Experience of a Tertiary Care Center. Saudi J Kidney DisTranspl 2011; 22: 1289-93. 

Athuraliya TNC, Abeysekera DTDJ, Amerasinghe PH, Kumarasiri PVR, Dissanayake V. Prevalence of chronic kidney disease in two tertiary care hospitals: high proportion of cases with uncertain aetiology. Ceylon Med J. 2009;54:23–5.

Katulanda P, Rathnapala DAV, Sheriff R, Mathews DR. Province and ethnic-specific prevalence of diabetes among Sri Lankan adults. Sri Lanka J Diabetes Endocrinol Metabolism. 2011;1:2–7)

Comment 24: It might be easier to describe this as a group of rhythm abnormalities – easier to read and understand

Response: Apart from rhythm abnormalities, we have assessed other cardiovascular morbidities like IHD from medical history/clinical records, hospitalization for congestive heart failure, soft tissue vascular calcification. Therefore, we changed the topic as “Prevalence, risk factors and predicted risk of cardiac events in Chronic Kidney Disease of uncertain aetiology in Sri Lanka: a tubular interstitial nephropathy”

Comment 25: Line 242 – important point – move up;

Response: Yes, it is important we moved up to top to the lines 255-258

Comment 26: Table 3 Best to avoid abbreviations as much as possible such as LVH, K, P, Na, Ca HCO3 – use the words instead

Response: Corrected in table 3 with using the words instead of abbreviations

Comment 27: Line 272 – instead of overall risk factor better to call it risk calculator or something like that

Response: Not only the variables from the risk calculator, we have analysed both traditional and non-traditional risk factors, medical history, lifestyle and most biochemical variables, which have probable relationship with cardiovascular disease. Therefore, we would like to keep that section topic as it is.

Comment 28: Start the paragraph with line 281 rather than the tables.

Response: Admitted the change in the lines 312-315.

Comment 29: My earlier comments about abbreviations apply to all the tables.

Response: Corrected the table using the words instead of abbreviations, where needed.

Comment 30: The discussion needs to re-written so it is focussed on the main findings of the study. Every finding of the study does not need to be discussed. The key finding is that patients with CKDu have lower risk fo CVD. There is no mention of the outcome of mortality anywhere and that would be useful to discuss and report on in this group. The first 3 paragraphs especially should be re-written.

Response: We have updated/rewritten the discussion section

Comment 31: Line 325 – is not was

Response: Corrected as “CKDu is a tubular interstitial disease among low socio-economic hardworking farmers in rural Sri Lanka” (lines 340-341)

Comment 32: Line 334 – How common is BMI>23 in Sri Lanka?

Response: Incidence of BMI>23 in Sri Lanka is vary from area to area. According to the Clinical Guidelines of the Endocrine Society of Sri Lanka, overweight and obesity was considered as BMI; 23 – 24.9 and BMI >25 respectively (Somasundaram et al., 2014). Based on that guidelines we have used BMI>23, which is considered to be overweight and have potential risk for cardiovascular disease.

(Somasundaram N, Rajaratnam H, Wijeyarathne C, Katulanda P, De Silva S, 469 Wickramasinghe, Clinical guidelines: The Endocrine Society of Sri Lanka; Management 470 of obesity. Sri Lanka J Diabetes. 2014;4:55–70)

Comment 33: Line 356 proteinuria is an important finding and should be given prominence

Response: Admitted that and explained more in the lines 375-380.

“ Proteinuria plays a significant role in the pathogenesis of IHD [46]. Nevertheless, in this study, the overall prevalence of proteinuria was as low as 26.05%. Hence, it was discovered that proteinuria was a late-stage finding in CKDu compared to early stages. Proteinuria has been associated with graded cardiovascular mortality, acting as risk multipliers across all levels of renal function in CKD (Herzog CA, 2009 and Nagata et al., 2013). Contrasting to CKD, proteinuria does not play a significant role in CVD pathogenesis in this specific disease of CKDu.”

(Herzog CA, Kidney disease in cardiology., Nephrol Dial Transplant. 2009 Jan; 24(1):34-7.

Nagata M, Ninomiya T, Kiyohara Y, et al. EPOCH-JAPAN Research Group Prediction of cardiovascular disease mortality by proteinuria and reduced kidney function: pooled analysis of 39,000 individuals from 7 cohort studies in Japan. Am J Epidemiol. 2013;178(1):1–11)

Comment 34: The limitations and strengths should be addressed openly.

Response: Admitted. Explained in the lines 419-428

Comment 35: Line 365 - please rewrite. I did not understand

Response: Edited the paragraph (in the lines 414-418)

---

## [Decision Letter · Decision Letter 1]

23 Dec 2020

PONE-D-20-19631R1

Prevalence, risk factors and predicted risk of cardiac events in Chronic Kidney Disease of uncertain aetiology in Sri Lanka: a tubular interstitial nephropathy

PLOS ONE

Dear Dr. Hettiarachchi,

Thank you for submitting your manuscript to PLOS ONE. After careful consideration, we feel that it has merit but does not fully meet PLOS ONE’s publication criteria as it currently stands. Therefore, we invite you to submit a revised version of the manuscript that addresses the points raised during the review process.

Please respond to the Reviewers comment regarding the rationale behind using PAHO/WHO criteria and the other edits requested. 

We look forward to receiving your revised manuscript.

Kind regards,

Rohina Joshi

Academic Editor

PLOS ONE

Reviewers' comments:

Reviewer's Responses to Questions

**Comments to the Author**

1. If the authors have adequately addressed your comments raised in a previous round of review and you feel that this manuscript is now acceptable for publication, you may indicate that here to bypass the “Comments to the Author” section, enter your conflict of interest statement in the “Confidential to Editor” section, and submit your "Accept" recommendation.

Reviewer #1: All comments have been addressed

Reviewer #2: All comments have been addressed

2. Is the manuscript technically sound, and do the data support the conclusions?

Reviewer #1: Yes

Reviewer #2: Yes

3. Has the statistical analysis been performed appropriately and rigorously? 

Reviewer #1: Yes

Reviewer #2: Yes

4. Have the authors made all data underlying the findings in their manuscript fully available?

Reviewer #1: Yes

Reviewer #2: (No Response)

5. Is the manuscript presented in an intelligible fashion and written in standard English?

Reviewer #1: Yes

Reviewer #2: (No Response)

6. Review Comments to the Author

Reviewer #1: Please include the rationale for using the PAHO/WHO (Pan American Health

Organization/World Health Organization cardiovascular) Risk Calculator over the other risk prediction tools.

Reviewer #2: Abstract

The sentence starting The cross-sectional study was conducted on patients with confirmed CKDu who were

attending two renal clinics in CKDu endemic-area. Consider re-writing to This cross-sectional study included patients with confirmed CKDu who were attending two renal clinics in CKDu endemic-area

The sentence starting – left ventricular hypertrophy is missing ‘in’

Line 98 – consider hypothesizing that the risk profile is different to CKD

Lby the study investigators.

How were patients invited?

Line 250 – please rewrite this sentence.

Line 334 – please rewrite the sentence – it has been reported.

Line 337 please rewrite the sentence – this lower prevalence of IHD in CKD

Line 354 – please rewrite the sentence – hence, it attributes to cardiovascular diseases

Line 360 – please rewrite the sentence – we found…

Line 381 – ECG is a widely ….

Line 382 – Ecgs were used not was.

7. PLOS authors have the option to publish the peer review history of their article (what does this mean?). If published, this will include your full peer review and any attached files.

Reviewer #1: No

Reviewer #2: **Yes: **Sradha S Kotwal

---

## [Author Response · Author response to Decision Letter 1]

26 Jan 2021

Index of Changes in Response to Reviewers’ Comments.

Title: Prevalence, risk factors and predicted risk of cardiac events in Chronic Kidney Disease of uncertain aetiology in Sri Lanka: a tubular interstitial nephropathy

Authors: Thilini W. Hettiarachchi1, Buddhi N.T.W.Fernando, Thilini Sudeshika, Zeid Badurdeen, Shuchi Anand, Ajith Kularatne, Sulochana Wijetunge, Hemalika T.K. Abeysundara, Nishantha Nanayakkara

Date: 25.01.2021

We thank the journal for the opportunity to address the reviewers’ comments, which will improve the paper. We have provided point-by-point responses to the reviewers’ comments below.

Reviewer #1: 

Comment 1: Please include the rationale for using the PAHO/WHO (Pan American Health Organization/World Health Organization cardiovascular) Risk Calculator over the other risk prediction tools.

Authors’ Response: The rationale has been described in our paper (see page 8, lines 163-169, page 20, lines 411 - 419 and page 21, lines 436 - 439). A validated cardiovascular risk prediction tool has not yet been implemented for CKDu; Sri Lanka. Herath et al in 2016 found that CVD risk assessment tools, both United Kingdom Prospective Diabetes Study risk engine and Framingham Risk Score have almost equal ability (former being marginally superior) in predicting underlying atherosclerotic vascular disease in patients with type 2 diabetes mellitus (T2DM) [58] . Nevertheless, in this study, we analysed a different population, who did not have a history of T2DM. Therefore, we used PAHO/WHO Risk Calculator, which is simple and easily implementable even in resource-poor settings with the readily available variables (gender, age, tobacco use, systolic blood pressure, diabetes and total cholesterol level) but with the limitation of not being validated for CKDu. Meanwhile, we agree a more appropriate validated risk prediction model is required for this particular group of patients for better results.

Reviewer #2: Abstract

Comment 1: The sentence starting The cross-sectional study was conducted on patients with confirmed CKDu who were attending two renal clinics in CKDu endemic-area. Consider re-writing to This cross-sectional study included patients with confirmed CKDu who were attending two renal clinics in CKDu endemic-area

Authors’ Response: We have revised the abstract and re-written the sentence line 35-36 “This cross-sectional study included patients with confirmed CKDu who were attending two renal clinics in CKDu endemic-area.”

Comment 2: The sentence starting – left ventricular hypertrophy is missing ‘in’

Authors’ Response: We have revised the manuscript and corrected the sentence as “Left ventricular hypertrophy was reported in 20.2% (n= 24)” (lines 45-46)

Comment 3: Line 98 – consider hypothesizing that the risk profile is different to CKD by the study investigators.

Authors’ Response: We have revised the manuscript and edited the lines 96 to 98 as “Understanding the importance, we designed this study to assess the burden of CVD, risk factors and the risk of developing CVD in CKDu, hypothesizing that the risk profile is different to CKD”.

Comment 4: How were patients invited?

Authors’ Response: We have revised the manuscript and included details on how patients invited for the study (see lines 105-108). The following sentences have now been incorporated to the manuscript to improve the clarity. Patients followed up in the routine renal clinics at Girandurukotte and Wilgamuwa hospitals were informed regarding the research in their clinic visit. Thereafter, all patients with the diagnosis of confirmed CKDu were invited to the study by medical officers of the clinic. 

Comment 5: Line 250 – please rewrite this sentence.

Authors’ Response: We have revised the manuscript and rewritten the sentence as” X-ray images showed soft-tissue vascular calcification in six (5.04%). Further, this vascular calcification was significantly associated (p=0.003, α= 0.05) with lower eGFR." (see lines 250-256)

Comment 6: Line 334 – please rewrite the sentence – it has been reported.

Authors’ Response: We have revised the manuscript and re-written the sentence as “At least 35% of patients with CKD, had evidence of an ischaemic event (myocardial infarction or angina) at the time of the presentation to a nephrologist.” (Lines 337-338)

Comment 7: Line 337 please rewrite the sentence – this lower prevalence of IHD in CKD

Authors’ Response: We have revised the manuscript and rewrote the sentence as “Furthermore, this finding is compatible with the characteristics of study population including physically active lifestyle, lower BMI, vegetable-based diet, lower prevalence of; diabetes, hypertension and proteinuria.” (Lines 340-342)

Comment 8: Line 354 – please rewrite the sentence – hence, it attributes to cardiovascular diseases

Authors’ Response: We have revised the manuscript and reworded the sentence as “Ultimately, the hyperuricemia attributes to cardiovascular diseases” (see lines 357-358)

Comment 9: Line 360 – please rewrite the sentence – we found…

Authors’ Response: We have revised the manuscript and rewritten the sentence as “Correspondingly, we observed that elevated Troponin-I was positively correlated with IHD in CKDu” (Lines 364-365)

Comment 10: Line 381 – ECG is a widely ….

Authors’ Response: We have revised the manuscript and corrected in the line 385

Comment 11: Line 382 – Ecgs were used not was.

Authors’ Response: We have revised the manuscript and corrected in the line 386 as “ECGs were used to identify rhythm abnormalities”

---

## [Decision Letter · Decision Letter 2]

22 Mar 2021

Prevalence, risk factors and predicted risk of cardiac events in Chronic Kidney Disease of uncertain aetiology in Sri Lanka: a tubular interstitial nephropathy

PONE-D-20-19631R2

Dear Dr. Hettiarachchi

We’re pleased to inform you that your manuscript has been judged scientifically suitable for publication and will be formally accepted for publication once it meets all outstanding technical requirements.

Kind regards,

Pasqual Barretti, Ph.D., MD

Academic Editor

PLOS ONE

Additional Editor Comments (optional):

Both reviewers have decided by the acceptation of the paper. I agree with them; in fact, the manuscript has improved importantly, since its first version.

Reviewers' comments:

Reviewer's Responses to Questions

**Comments to the Author**

1. If the authors have adequately addressed your comments raised in a previous round of review and you feel that this manuscript is now acceptable for publication, you may indicate that here to bypass the “Comments to the Author” section, enter your conflict of interest statement in the “Confidential to Editor” section, and submit your "Accept" recommendation.

Reviewer #1: All comments have been addressed

Reviewer #2: All comments have been addressed

2. Is the manuscript technically sound, and do the data support the conclusions?

Reviewer #1: Yes

Reviewer #2: Yes

3. Has the statistical analysis been performed appropriately and rigorously? 

Reviewer #1: Yes

Reviewer #2: Yes

4. Have the authors made all data underlying the findings in their manuscript fully available?

Reviewer #1: Yes

Reviewer #2: No

5. Is the manuscript presented in an intelligible fashion and written in standard English?

Reviewer #1: Yes

Reviewer #2: Yes

6. Review Comments to the Author

Reviewer #1: Authors have addressed the clarifications sought. The manuscript covers the importance of accessing cardiovascular risk in CKDu.

Reviewer #2: Introduction

Line 58 – multiple studies around the globe have reported…

Line 62 – at instead of as

Line 71 re write to – Novel risk factors of inflammation, bone and mineral disorders, hyperphosphatemia, hypercalcemia, secondary hyperparathyroidism, and oxidative stress, all of which are attributed to compromised renal function, are highly associated with elevated cardiovascular risk in patients with kidney disease

Line 90 – rates of inadvertent exposure….

Line 93 have not rather than were not

Line 108 – rewrite to - written informed consent was obtained from all recruited participants.

Line 166 and its use of readily available.

Line 169 – the following parameters

Line 250 a normal pericardium

Table 4 – the p values should be to 2 decimal places unless they are 0.001

All other values can also be to 2 decimal places.

Avoid abbreviations.

Line 310 denoting rathe than denoted.

Table 5 – comments as for table 4

Line 350 add years after >50

7. PLOS authors have the option to publish the peer review history of their article (what does this mean?). If published, this will include your full peer review and any attached files.

Reviewer #1: No

Reviewer #2: **Yes: **Sradha Kotwal

---

## [Editor Report · Acceptance letter]

31 Mar 2021

PONE-D-20-19631R2 

Prevalence, risk factors and predicted risk of cardiac events in Chronic Kidney Disease of uncertain aetiology in Sri Lanka: a tubular interstitial nephropathy 

Dear Dr. Hettiarachchi:

I'm pleased to inform you that your manuscript has been deemed suitable for publication in PLOS ONE. Congratulations! Your manuscript is now with our production department. 

Kind regards, 

on behalf of

Prof. Pasqual Barretti 

Academic Editor

PLOS ONE